# Using structured pathology data to predict hospital-wide mortality at admission

Mieke Deschepper[1]*, Willem Waegeman[2], Dirk Vogelaers[3,4], Kristof Eeckloo[1,5]

**1** Strategic Policy Cell at Ghent University Hospital, Ghent, Belgium, **2** Department of Data Analysis and Mathematical Modelling, Ghent University, Ghent, Belgium, **3** General Internal Medicine, Ghent University Hospital, Ghent, Belgium, **4** Dept. of Internal Medicine, Ghent University, Ghent, Belgium, **5** Department of Public Health and Primary Care, Ghent University, Ghent, Belgium

* Mieke.deschepper@uzgent.be

**Data Availability Statement:** Data have been uploaded to GitHub and are accessible using the following link: https://github.com/descheppermieke/Using-structured-pathology-

## Abstract

Early prediction of in-hospital mortality can improve patient outcome. Current prediction models for in-hospital mortality focus mainly on specific pathologies. Structured pathology data is hospital-wide readily available and is primarily used for e.g. financing purposes. We aim to build a predictive model at admission using the International Classification of Diseases (ICD) codes as predictors and investigate the effect of the self-evident DNR ("Do Not Resuscitate") diagnosis codes and palliative care codes. We compare the models using ICD-10-CM codes with Risk of Mortality (RoM) and Charlson Comorbidity Index (CCI) as predictors using the Random Forests modeling approach. We use the Present on Admission flag to distinguish which diagnoses are present on admission. The study is performed in a single center (Ghent University Hospital) with the inclusion of 36 368 patients, all discharged in 2017. Our model at admission using ICD-10-CM codes (AUCROC = 0.9477) outperforms the model using RoM (AUCROC = 0.8797 and CCI (AUCROC = 0.7435). We confirmed that DNR and palliative care codes have a strong impact on the model resulting in a decrease of 7% for the ICD model (AUCROC = 0.8791) at admission. We therefore conclude that a model with a sufficient predictive performance can be derived from structured pathology data, and if real-time available, can serve as a prerequisite to develop a practical clinical decision support system for physicians.

## 1. Introduction

### 1.1 Reuse of readily available hospital-wide data

Large amounts of data are registered in well-defined formats in hospitals. These datasets contain administrative data—such as age, billing data, specialism, and so on—and structured pathology data using the International Classification of Diseases (ICD) codes. Such datasets exhibit much information that should be useful for secondary goals, although this information is currently unused for predicting in-hospital mortality. Added value could be generated from existing hospital databases without the need for much additional effort or time being spent on noncare activities on the part of caregivers.

data-to-predict-hospital-wide-mortality-at-
admission.

**Funding:** For this work Willem Waegeman received
funding from the Flemish Government under the
"Onderzoeksprogramma Artificiële Intelligentie (AI)
Vlaanderen" programme.

**Competing interests:** The authors have declared
that no competing interests exist.

## 1.2 Current approach predicting mortality

Early identification of patients with a high risk of mortality is crucial to adequately and timely
act by health care providers. The prediction of mortality is a well-researched topic in intensive
care [1–3] and cardiac diseases [4], yet little research has been based on hospital-wide datasets.
Currently, the Charlson Comorbidity Index (CCI) and the Risk of Mortality (RoM) score are
widely used to predict mortality.

The CCI score is obtained from 17 weighted comorbidities and was initially developed to
assess risk of one-year mortality [5]. This method dates from 1987 and is intended to provide a
fast and easy risk assessment. The clinical conditions were initially retrieved manually from
hospital charts, but are now available as ICD-10-CM codes, allowing automated extraction
and calculation of the score in larger samples, as has been done by Quan (2005) and Sundarar-
ajan (2004) [6, 7]. Although the initial purpose of this measure is to asses risk of one year mor-
tality, as this measure is still used to predict in-hospital mortality, we add this to our list of
comparison measures.

RoM represents the likelihood of dying calculated from all comorbidities [8] based on all
ICD codes: a weight is given to all secondary diagnoses. In the second step, the standard risk of
mortality level of each secondary diagnosis is modified based on patient age, principal diagno-
sis, pathology group, and procedures. This is aggregated into subclasses, numbered 1 to 4, rep-
resenting categories rather than scores. RoM should not be confused with Severity of Illness
(SoI) score, which is calculated from the same data. SoI is defined as the extent of organ system
derangement or physiologic decompensation [8]. RoM is used for risk adjustment of in-hospi-
tal mortality indicators from the Agency for Healthcare Research and Quality (AHRQ). This
categorization of risk of mortality has previously been demonstrated to correlate strongly with
observed mortality in a medical ICU setting [9]. The algorithm to calculate RoM is neither free
nor open source, and as such a license is needed. RoM was shown to have a better predictive
value than CCI for in-hospital mortality, but this study only encompassed older patients in
surgical settings [10]. Furthermore, separate diagnoses of CCI scores have been demonstrated
to better predict in-hospital mortality than the score itself in hip fracture patients [11].

Many hospitals make structured pathology information available in the form of ICD-
10-CM codes. In many countries, including Belgium, this classification is obligated to calculate
the hospital reimbursement. The ICD-10-CM diagnosis code is a seven character code, which
can be approached as a chapter (one character), a category (three characters) and a full code
(all seven characters; see Fig 1). These codes form the basis of the aggregated pathology groups
(APR-DRG) and the RoM and SoI values.

In Belgian hospitals, these codes are obtained via an extensive manual process. Trained
ICD-coding-experts search for the pathology described for a patient in the Electronic Health
Record (EHR) (and other databases or manual records), with the discharge letter being one of
the main sources. They then translate the patient's diagnoses and procedures at admission into
adequate ICD codes.

## 1.3 Proposed approach predicting in-hospital mortality

RoM and CCI are both aggregated measures with ICD-10-CM codes as their basis. The first
difference in approach is to use the individual ICD-10-CM diagnosis codes as predictors
instead of these aggregated measures. In order to develop an early-warning system for in-hos-
pital mortality that would be useful in practice, it is important to concentrate on variables
known at admission. However, most studies considering ICD diagnosis (or aggregations like
RoM) as predictors for in-hospital mortality, make use of all the codes registered upon comple-
tion of the hospitalization episode, including those generated by complications during

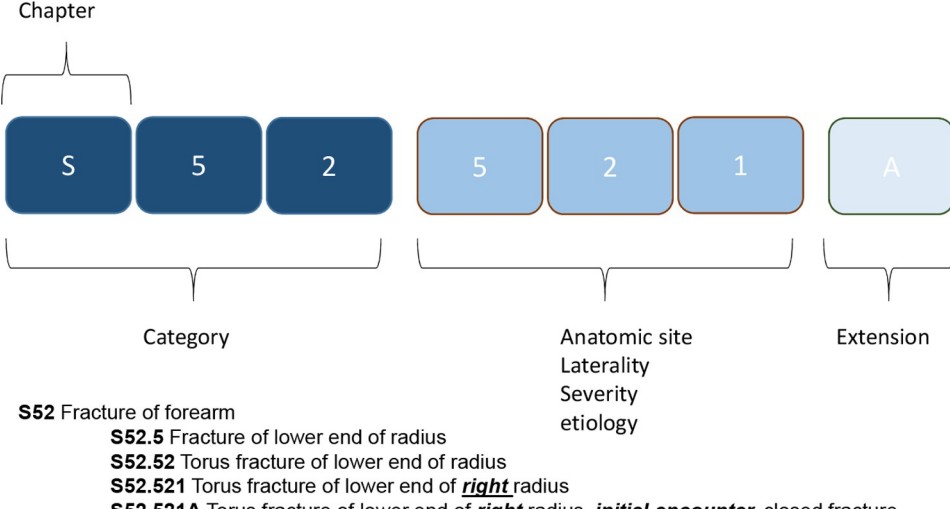

**Fig 1. ICD-10-CM diagnosis code hierarchy and with example: S52, fracture of the forearm.** The ICD-10-CM code consists of a chapter (S), category (S52) and full code (S52.521A).

hospitalization. To address this problem, the "Present on Admission" (PoA) flag can be used to indicate that a diagnosis was present at the time the admission order was written. This flag is available in some ICD-9-CM databases, but is mandatory from ICD-10-CM onwards. Adding this flag results in better models for in-hospital mortality prediction [12]. Next, nearly all studies that build a prediction model on ICD data have been restricted in scope to specific pathologies, with the exception of two hospital-wide studies [13, 14]. These include in their analysis all diagnoses and procedures at the moment of discharge. Likewise, in order to be practically useful in a real world setting, RoM should also be calculated at admission, rather than on the complete ICD-10-CM coding set after completion of the hospital episode. We are not aware of any studies that have looked at the prediction of in-hospital mortality using ICD codes or RoM categories at admission.

Patients with a "Do Not Resuscitate" (DNR) or a palliative care code at admission are already at high (and intrinsically predictable) risk of mortality. We did not find any articles in the literature that excluded these patients from a hospital-wide model predicting in-hospital mortality. We hypothesize that these diagnoses are of limited relevance in the development of an in-hospital mortality warning system. As such, we aimed to build a model excluding these patients.

A final difference from earlier research is the hospital-wide scope of the model and the application of machine learning (ML) techniques. In recent studies ML techniques have been adopted instead of the more commonly used logistic regression (LR), as ML techniques perform better at prediction than LR [15, 16].

## 1.4 Objective

The objective of this study is to assess whether in-hospital mortality can be predicted accurately through individual ICD-10-CM codes available at admission, and to compare and evaluate this approach with existing scoring systems based on CCI and RoM. Our analysis quantifies the performance of aggregated CCI and RoM scores versus individual ICD-10-CM codes on a large hospital-wide group of pathologies, excluding DNR and palliative care codes.

This may lead to a data-driven, machine learning approach based on nonlinear models, resulting in a clinical decision support system.

## 2. Materials and methods

### 2.1 Study population and study variables

The study cohort includes patients discharged between 1 January 2017 and 31 December 2017 at a single center, the 1061-bed University Hospital, Ghent, Belgium. Hospitalized patients were excluded from the analysis if they lacked detailed coding due to incomplete records, and also in the case of particular patient groups subject to specific hospital budget rules. This mainly refers to patients who stayed for more than half of their hospitalization period in a psychiatric department; no ICD coding available for these.

The dataset at admission contains only diagnoses positively flagged as PoA and diagnoses that are always present on admission, such as 'Z880 Allergy status to penicillin', 'Z794 Long term (current) use of insulin' or 'I252 Old myocardial infarction' [17]. As we are hypothesizing that DNR and palliative care codes at admission have a high but essentially unnecessary impact for the development of an in-hospital mortality warning system, and since the prediction for such patients would not be contributive, we also fit models omitting patients with these codes. In order to confirm or invalidate the performance of the predictors, models are also fitted at discharge (with all diagnosis and measures calculated at discharge). One of the differences between RoM and ICD is the principal diagnosis: we also create datasets without a principal diagnosis in order to gain insight into its weight on the models.

The measure for RoM and CCI are recalculated on all datasets. The calculation of CCI is straightforward, for RoM we use the 3M algorithm under the license of Ghent Univerity Hospital.

The dependent (outcome) variable is in-hospital mortality. RoM, CCI or ICD-10-CM diagnoses are taken as predictors. RoM is added as an ordinal variable, CCI as a continuous score and in models with an ICD-10-CM diagnosis the three hierarchy levels (chapter, category, code) are all added as dummy variables, translating into flags in the column for chapter 20 (S), a flag for category S52 and a flag for code S52.521A in the example given in Fig 1.

We only include the diagnosis codes in our ICD models, not the procedure codes. For planned admission, we can assume that a single surgical procedure was the reason for hospitalization, but this certainty is not possible when multiple procedures are performed, as our strategy is to minimize the number of false positives.

### 2.2 Statistical analysis

We use the Random Forests approach to build the predictive model. It is known that this nonlinear method outperforms logistic regression, which is more commonly used in medical applications [18]. In comparison with other data-driven approaches, Random Forests tends to perform as one of the best techniques overall for solving classification problems [3, 4]. Furthermore, on a similar dataset with unplanned readmissions as the outcome variable, Random Forests turned out to outperform penalized logistic regression and the gradient boosting machine learning approach [19]. The existing literature was scrutinized to determine which methods have already been proven to deliver solid solutions. We do not use deep learning methods, which are very popular, and have been used in recent research [20, 21] dealing with similar outcomes. Such techniques are very suitable for complex features, such as pixels in images, but they are not useful for standard tabular datasets as in this study.

For all models, the data is first split into training (60%), validation (20%), and test (20%) sets, in order to prevent overfitting. Otherwise models that can simply "remember" the

training data (rather than generalizing from it) would be rewarded. The two main parameters for the model are the number of trees (fixed here at one thousand) and the maximum depth, which is tuned. The model can be interpreted using a variable importance plot. An implementation as described in [22] is used.

We compare four sets of predictors: CCI, RoM, ICD and ICD_noPDX. The last of these, ICD_noPDX, is the set of ICD diagnoses without the principal diagnosis. This is defined as a separate category in order to assess whether the principal diagnosis impacts the models, as RoM is calculated on the basis of all comorbidities without this principal diagnosis. For each set of predictors, we calculate and compare three models: 1) at admission, 2) at admission excluding patients with DNR or palliative care diagnosis code, and 3) at discharge.

We assess the performance of each model with the Area Under the Receiver Operating Characteristic curve (AUCROC) on the test dataset. The AUCROC is typically preferred over other measures in situations where the data is imbalanced as in our study and other health care datasets [23]. We also calculate the Area Under the Precision-Recall Curve (AUCPR), which mainly focusses on correctly predicting patients for which the model assumes they have a high probability of mortality [24]. The ROC curve shows the False Positive Rate on the $x$-axis and the True Positive Rate on the $y$-axis, while the PR curve has the True Positive Rate (or Recall) on the $x$-axis and the Precision (or positive predictive value) on the y-axis.

All analyses are performed using the R Statistical Software, version 3.4.1 with the h2o and mltools packages.

The study was approved by the ethics committee at Ghent University Hospital (Belgian registration no. B670201836838).

## 3. Results

### 3.1 Description of the study variables

A total of 36 368 patients were hospitalized and discharged during the study period. After excluding admissions as per protocol, the final study cohort included 34 671 patients, of whom 919 (3%) did not survive. 41% of the included patients belonged to a surgical pathology group. The excluded patients were all admissions in the psychiatry department except for three with incomplete records. 1063 patients had a DNR or palliative care code at admission; after excluding these, 33 608 patients remained in the cohort that was modeled.

Table 1 provides an overview of the characteristics of the survivors and non-survivors, adding age and sex for demographic description (these were not used in the models). For continuous variables we show the median with the first and third quartiles. CCI scores and RoM categories are shown upon admission and discharge. The CCI scores do not differ, but the distribution for RoM categories for non-survivors at discharge differs from the distribution at admission.

### 3.2 Models

The resulting AUCs derived from our models are summarized in Table 2, using the four predictor sets at admission, with and without excluding patients with DNR or palliative care diagnosis code, and at discharge. We also add the number of predictors for each set (row) and the number of records included per model type (column).

The models using ICD-10-CM codes as predictors outperform the others. The models using CCI as predictors have low AUCROC. The resulting ROC curves are shown in Fig 2A, while the resulting Precision-Recall plots are shown in Fig 2B.

The difference between the models using ICD-10-CM as predictors and RoM is the smallest when excluding DNR and palliative care codes. When we exclude the principal diagnosis from

**Table 1. Population overview: Characteristics of survivors and non-survivors.**

| Population overview | | survivors (N: 33 752 = 97%) | non-survivors (N: 919 = 3%) | p-value* |
|---|---|---|---|---|
| Age | | 52 [30–67] | 70 [58–80] | <0.001 |
| Sex (% male) | | 17 320 (51%) | 543 (59%) | <0.001 |
| % Diagnoses Present on admission (PoA) | | 93% | 80% | <0.001 |
| DNR at admission | | 469 (1.5%) | 231 (25%) | <0.001 |
| Palliative care flag at admission | | 234 (0.7%) | 286 (31%) | <0.001 |
| CCI | Admission | 0 [0–2] | 3 [1–6] | <0.001 |
| | Discharge | 0 [0–2] | 3 [1–6] | <0.001 |
| RoM (1–2–3–4) | Admission | 71% - 22% - 6% - 1% | 10% - 34% - 40% - 16% | <0.001 |
| | Discharge | 70% - 22% - 7% - 1% | 6% - 22% - 40% - 33% | <0.001 |

Data are reported as n (%) or medians (1$^{st}$– 3$^{rd}$ quartile), or otherwise indicated.

* p-values based on Pearson chi-square for categorical variables and the Wilcoxon rank-sum test for continuous variables.

Legend: DNR = Do Not Resuscitate; PoA = Present on Admission flag; CCI = Charlson Comorbidity Index; RoM = Risk of Mortality

the ICD predictor set, the set still delivers a better prediction than using the aggregated RoM. As hypothesized, the models excluding patients with DNR and palliative code at admission have lower AUCROC and AUCPR. The effect of these diagnosis codes is visually shown in the variable importance plot (Fig 3). The most important variables for each model are shown next to each other. Due to the imbalanced dataset, the difference between the results using AUCPR is even more pronounced in favor of using ICD as predictor set.

## 4. Discussion

This study establishes that modeling based on all individual ICD-10-CM codes is a better predictor of in-hospital mortality at admission than hitherto used combined scores such as RoM and CCI. The PoA flag available in ICD-10-CM is necessary in order to retain only those diagnoses recognized at admission in the model. RoM does not allow the automatic exclusion of diagnoses not present upon admission, in contrast to ICD coding, which favors the latter in the development of a clinically-relevant decision support system. From Table 1, we could already indicate that for non-survivors the diagnoses not present at admission have an impact

**Table 2. Model AUC results.**

| | | Admission | | | | Discharge | | |
|---|---|---|---|---|---|---|---|---|
| | # predictors | All (n = 34 671) | | Excluding patients with DNR or palliative care diagnosis code (n = 33 608) | | # predictors | All (n = 34 671) | |
| | | AUCROC | AUCPR | AUCROC | AUCPR | | AUCROC | AUCPR |
| CCI | 1 | 0.7435 | 0.0615 | 0.7015 | 0.0270 | 1 | 0.7471 | 0.0654 |
| RoM | 4 | 0.8797 | 0.1393 | 0.8601 | 0.1086 | 4 | 0.9272 | 0.1979 |
| ICD | 4743 | 0.9477 | 0.4035 | 0.8791 | 0.2476 | 4961 | 0.9774 | 0.5542 |
| ICD_noPDX | 3761 | 0.9340 | 0.3837 | 0.8623 | 0.1911 | 4050 | 0.9671 | 0.5425 |

AUC results for the Random Forests models: each line is a set of predictors. At admission we built the model for all diagnoses and excluding admissions with DNR or palliative care code. At discharge all diagnoses known for the whole episode were used in the models. As the dataset is imbalanced, the AUCPR is shown as well as the AUCROC. The differences between the predictor sets are larger using AUCPR.

Legend: CCI = Charlson Comorbidity Index; RoM = Risk of Mortality; ICD = International Classification of Diseases; DNR = Do Not Resuscitate; ICD_noPDX = all ICD codes without the principal diagnosis code; AUCROC = Area Under the ROC curve; AUCPR = Area under Precision-Recall Curve

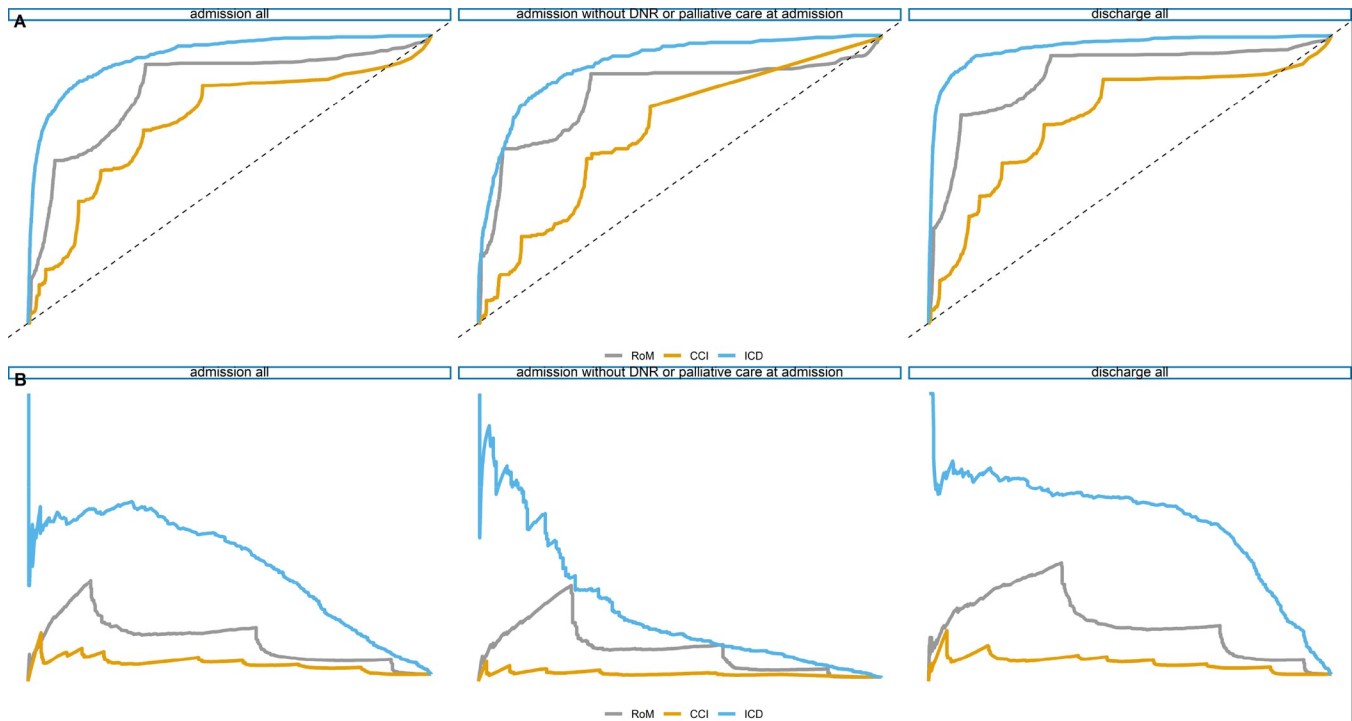

**Fig 2.** A (Upper panel). ROC curves showing the results using three different sets of predictors in the Random Forests model. The figure on the left shows the ROC curve with all diagnoses known at admission, while the figure on the right shows all diagnoses known at discharge. The ROC curve in the middle is for the models using only the diagnoses known at admission, excluding all admissions with DNR and palliative care codes at admission. B (Lower panel). Precision-Recall plot showing the results using three different sets of predictors in the Random Forests model. The figure on the left shows the PR curve with all diagnoses known at admission and on the right all diagnoses known at discharge. The PR curve in the middle are the models using only the diagnoses known at admission, excluding all admissions with codes DNR or palliative care codes at admission. Both approaches show low performance using CCI as a predictor for in-hospital mortality, while the models using ICD as predictors perform best overall. Legend: CCI = Charlson Comorbidity Index; RoM = Risk of Mortality; ICD = International Classification of Diseases.

on the RoM classification. Evidently and intrinsically, the codes for DNR and palliative care—as key components of care paths—aimed at humanizing the dying process. They avoid futile care, which is very strongly associated with subsequent in-hospital mortality. Hence they clearly need to be excluded from prediction models of in-hospital mortality at admission, as these models should be aimed at providing a relevant support tool for clinical decision making.

We confirm that individual diagnoses perform better than the aggregated measures of the CCI score [11]. Our results show that CCI cannot be considered as a robust predictor for in-hospital mortality and should not be further used for this purpose. Moreover, prediction of in-hospital mortality was not the initial purpose of introducing CCI, which was rather developed to assess mortality after one year.

RoM turns out to be suboptimal compared to the full ICD diagnosis set. We have to be aware that RoM is calculated individually, belonging to a certain pathology group (APR-DRG). As such, this measure should be considered at the individual level or by pathology group. For the prediction models this requirement is fulfilled, as the patients are handled individually. One advantage of RoM compared to all the ICD codes is that is has very easy and intuitive properties. Also, the restriction to only four predictors (instead of almost 5000) saves computation time in building the model. However, an extra calculation step is needed to retrieve the RoM at admission, which is not a common practice. This also requires that the necessary 3M

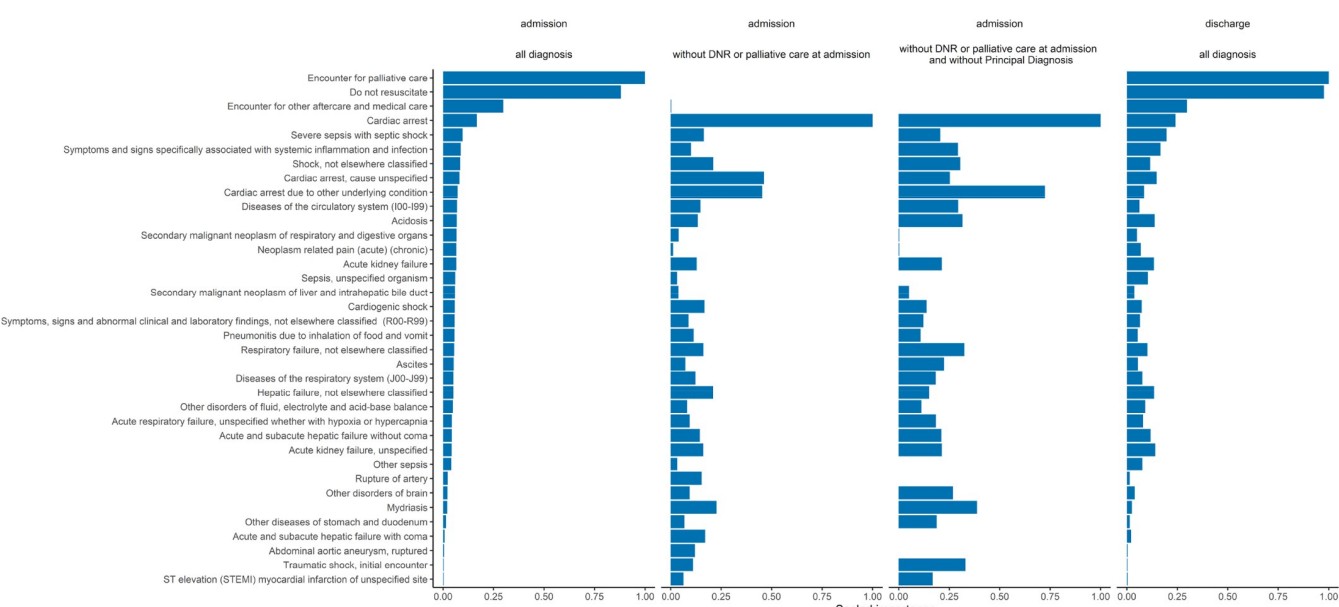

**Fig 3. Variable importance plot for the models using ICD-10-CM diagnosis codes as predictors at admission and discharge, either using all diagnosis codes, without the *Do Not Resuscitate* or *palliative care* codes at admission, or without these codes and without the principal diagnosis.** Whereas 'Encounter for palliative care', 'Do not resuscitate' and 'Encounter for other aftercare and medical care' are the three most important variables in the models with all diagnosis, 'Cardiac arrest' is the most important variable in the model with the excluded patients.

license is available in the software package (e.g., EHR), as the calculation is needed at real time and may come with an extra license cost. One of the key differences between the aggregated RoM score and the ICD predictors is the inclusion of the principal diagnosis. Where RoM applies this for risk adjustment, our models with ICD codes uses this principal diagnosis function as a full predictor. The models we constructed excluding the principal diagnosis as predictor all perform better than the models based on RoM. Another potential explanation for the difference could be the fixed weighting in the RoM calculation; this is unlike the weighting of the predictors, which depends in our ICD models on the patient mix used for training.

We have found only a few prior hospital-wide studies predicting in-hospital mortality at admission in the literature [20]. One study using deep learning as technique and extracting the data from EHRs into a specific format achieved an AUCROC of 0.90. The potential of ICD-10-CM diagnoses as predictors in hospital-wide studies has not been sufficiently researched and as such we can only compare the AUCROC calculated from our model to that based on RoM at moment of discharge on all patients (RoM_AUCROC = 0.93). In this perspective, a single study only included non-cardiac surgery patients in its model [10], obtaining an AUCROC of 0.97. In another study including only non-chirurgical patients [14], an AUCROC of 0.86 was observed. In a preliminary study [13] on the same data set, without the inclusion of laboratory data and without using a penalization factor in the logistic regression, an AUCROC of only 0.81 was achieved. As other techniques, patient mix, and predictors are included, the models are not fully comparable, hampering conclusions. Our models containing all ICD-10-CM diagnosis codes at discharge already have an AUCROC of 0.98, and could be optimized using variables such as age. We believe that our models have a good performance with still a large potential for improvement.

In our study, the codes are manual retrieved by ICD-coding-experts using the information from the EHR. This implies a certain degree of human error and bias within the codes. These codes, however are also the basis for the calculation of CCI and RoM and as such the same bias

holds for all predictors. We are also aware that the codes for DNR will be biased: there will be a general tendency towards undercoding rather than overcoding, as they do not influence the severity of illness of an individual patient and as such do not have an impact on the financial reimbursement. However, our results show that patients with these codes already have a large effect on the model. As such, we believe that the potentially-missing codes will only have a minor effect on our models. The same holds for the PoA flag. This flag will be biased, as human interference is needed to unflag the diagnosis code. However, the results comparing the measures will remain the same, as the same codes are flagged for all measures. The results of the models may be slightly biased due to repeated measures (e.g. if a patient was admitted more than one time during the study period), as all admissions are treated equally. To overcome this bias, we should remove all admissions previous to the readmissions. In our dataset only 2.5% of the admissions are readmitted within 30 days.

To optimize the model we should include administrative variables with proven importance for adjusting mortality risk [25]. We did not include the ICD-10-PCS procedures codes, as we could not distinguish which procedure was known and planned upon admission. RoM implicitly uses procedures, as the RoM is risk-adjusted for the procedures and depends on a pathology group (which may be non-surgical or surgical). Nevertheless, our models with ICD-10-CM diagnosis codes still outperform RoM as a predictor, and could thus only be improved. The predictive performance of the risk adjustment models could be further improved, among other factors, through the inclusion of laboratory data, as shown in many studies [12, 14, 18, 26]. It has been shown that a limited set of routine laboratory results upon admission can contribute to risk stratification and independently predict mortality in patients hospitalized with acute heart failure [27]; the inclusion of laboratory data at admission from our EHR thus seems necessary [28]. In many electronic health records these laboratory data can be found as Logical Observation Identifiers Names and Codes (http://www.regenstrief.org/resources/loinc/).

The dataset for this study was extracted from a single center. It is possible that the performance may differ in other institutions with other patient mixes. However, we believe that the conclusions should be independent of the actual patient mix. We should also be aware that in some cases no represented sample is available in our historical data. Not only will there be new patients, with different case mixes, but ICD-10-CM also has yearly updates with the introduction of new codes. Refreshing of the model on a frequent basis is thus necessary in order to continue optimal predictions.

In conclusion, a predictive model with trustworthy operating characteristics can be derived from compulsory administrative data. A predictive model containing ICD-10-CM codes outperforms the conventional tools of combined scores. Data available at admission are required to develop a clinically-relevant warning system. An automated system would allow real-time alerts, without increasing workload and additional costs, while improving patient outcomes.

## Author Contributions

**Conceptualization:** Mieke Deschepper, Willem Waegeman, Dirk Vogelaers, Kristof Eeckloo.

**Data curation:** Mieke Deschepper.

**Formal analysis:** Mieke Deschepper.

**Methodology:** Mieke Deschepper, Willem Waegeman.

**Project administration:** Mieke Deschepper.

**Validation:** Willem Waegeman.

**Visualization:** Mieke Deschepper.

**Writing – original draft:** Mieke Deschepper.

**Writing – review & editing:** Mieke Deschepper, Willem Waegeman, Dirk Vogelaers, Kristof Eeckloo.

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
