## [Decision Letter · Decision Letter 0]

27 May 2020

PONE-D-20-08968

Using structured pathology data to predict hospital-wide mortality at admission

PLOS ONE

Dear Dr. Deschepper,

Thank you for submitting your manuscript to PLOS ONE. After careful consideration, we feel that it has merit but does not fully meet PLOS ONE’s publication criteria as it currently stands. Therefore, we invite you to submit a revised version of the manuscript that addresses the points raised during the review process.

This is an interesting manuscript. Overall, the methods of the study are appropriate, the results are clearly presented and the discussion is well developed. However, there are some points that should be addressed.

First, the authors must respond to the questions raised by the reviewer.I have other minor comments. Figure 3 presents the importance of variables for three statistical models. It would be interesting to add a fourth column showing the importance of diagnoses for the ICD_noPDX model (all the diagnoses except the principal one) excluding admissions with DNR or palliative care code. Also, in table1 the row “Present on admission (%)” needs to be clarified. I suppose that are percentages of diagnoses, but it is somehow confuse because the columns are referred to patients instead of diagnoses.In addition, the PLOS Data policy requires authors to make all data underlying the findings described in their manuscript fully available without restriction, with rare exception (https://journals.plos.org/plosone/s/data-availability). If there are ethical or legal restrictions on sharing a sensitive data set, authors should provide further information within their Data Availability Statement

We look forward to receiving your revised manuscript.

Kind regards,

Juan F. Orueta, MD, PhD

Academic Editor

PLOS ONE

2. Please ensure that your method is described in sufficient detail to meet our criteria on reproducibility http://journals.plos.org/plosone/s/submission-guidelines#loc-methods-software-databases-and-tools.

3. In ethics statement in the manuscript and in the online submission form, please provide additional information about the patient records used in your retrospective study. Specifically, please ensure that you have discussed whether all data were fully anonymized before you accessed them and/or whether the IRB or ethics committee waived the requirement for informed consent. If patients provided informed written consent to have data from their medical records used in research, please include this information.

5. Your ethics statement must appear in the Methods section of your manuscript. If your ethics statement is written in any section besides the Methods, please move it to the Methods section and delete it from any other section. Please also ensure that your ethics statement is included in your manuscript, as the ethics section of your online submission will not be published alongside your manuscript.

Reviewers' comments:

Reviewer's Responses to Questions

**Comments to the Author**

1. Is the manuscript technically sound, and do the data support the conclusions?

Reviewer #1: Yes

Reviewer #2: Yes

2. Has the statistical analysis been performed appropriately and rigorously? 

Reviewer #1: Yes

Reviewer #2: Yes

3. Have the authors made all data underlying the findings in their manuscript fully available?

Reviewer #1: Yes

Reviewer #2: Yes

4. Is the manuscript presented in an intelligible fashion and written in standard English?

Reviewer #1: Yes

Reviewer #2: Yes

5. Review Comments to the Author

Reviewer #1: This is an interesting and well performed study on the prediction of in-hospital mortality using ICD-10-CM codes. The authors used a retrospective data from patients admitted to a single center in Belgium during one year. The study uses Random Forests approach to build the prediction model. Based on the above data set, a comprehensive model for the prediction of in-hospital mortality was devised. The authors show that their model performs better than RoM and CCI. While the study has merit there are few issues that need to be addressed.

The dataset for hospitalized patients may include repeated measure data (e.g., if a patient was admitted than one time during the study period). Some patients (roughly 12-15% of patients are readmitted within one month and perhaps ~ 25% are readmitted within 3 months). The authors should address this issue (how this was adjusted for). Only one observation should be obtained per patient, and if not, that should be addressed or discussed.

Minor comments;

1. Please add p values to Table 1.

Reviewer #2: There are many studies on mortality prediction in the hospital, and this study is very interesting. The number of cases is 30,000 or more, which is a large-scale data, which is very useful in data analysis. In addition, it is novel that the analysis excludes DNR cases, and we believe that this study can provide scientifically important information.

6. PLOS authors have the option to publish the peer review history of their article (what does this mean?). If published, this will include your full peer review and any attached files.

Reviewer #1: No

Reviewer #2: No

---

## [Author Response · Author response to Decision Letter 0]

3 Jun 2020

Comments from the editor and reviewers

Editor

I have other minor comments. Figure 3 presents the importance of variables for three statistical models. It would be interesting to add a fourth column showing the importance of diagnoses for the ICD_noPDX model (all the diagnoses except the principal one) excluding admissions with DNR or palliative care code. 

Response: We updated Figure 3 and added the extra column with the model at admission without the principal diagnosis and without the admissions with DNR or palliative care codes at admission.

Figure 3 Variable importance plot for the models using ICD-10-CM diagnosis codes as predictors at admission and discharge, either using all diagnosis codes, without the Do Not Resuscitate or Palliative care codes at admission, or without these codes and without the Principal Diagnosis. Whereas ‘Encouter for palliative care’, ‘Do not resuscitate’ and ‘Encouter for other aftercare and medical care’ are the three most important variables in the models with all diagnosis, ‘Cardiac arrest’ is the most important variable in the model with the excluded patients.

Also, in table1 the row “Present on admission (%)” needs to be clarified. I suppose that are percentages of diagnoses, but it is somehow confuse because the columns are referred to patients instead of diagnoses.

Response: The interpretation of the row is indeed the percentage of diagnoses. We acknowledge that the description is not clear and changed this to “% Diagnoses Present on admission (PoA)”. 

In addition, the PLOS Data policy requires authors to make all data underlying the findings described in their manuscript fully available without restriction, with rare exception (https://journals.plos.org/plosone/s/data-availability). If there are ethical or legal restrictions on sharing a sensitive data set, authors should provide further information within their Data Availability Statement

Response: We have reviewed the ethical and legal restrictions regarding the data. As this contains no unique identifier we can anonymize the data and make the data publicly available.

We made a github repository with all datasets and a description in the README file. The link of the repository: https://github.com/descheppermieke/Using-structured-pathology-data-to-predict-hospital-wide-mortality-at-admission

Reviewer #1

This is an interesting and well performed study on the prediction of in-hospital mortality using ICD-10-CM codes. The authors used a retrospective data from patients admitted to a single center in Belgium during one year. The study uses Random Forests approach to build the prediction model. Based on the above data set, a comprehensive model for the prediction of in-hospital mortality was devised. The authors show that their model performs better than RoM and CCI. 

Response: We thank the reviewer for the kind words and for the time spent to improve our study.

While the study has merit there are few issues that need to be addressed.

The dataset for hospitalized patients may include repeated measure data (e.g., if a patient was admitted than one time during the study period). Some patients (roughly 12-15% of patients are readmitted within one month and perhaps ~ 25% are readmitted within 3 months). The authors should address this issue (how this was adjusted for). Only one observation should be obtained per patient, and if not, that should be addressed or discussed.

Response: We thank the reviewer for this comment. We acknowledge that there is a bias due to some readmissions and will add this to the Discussion. We do believe that the bias is limited due to a low readmission rate (< 3% Deschepper M, Eeckloo K, Vogelaers D, Waegeman W. A hospital wide predictive model for unplanned readmission using hierarchical ICD data. Comput Methods Programs Biomed. 2019. Epub 2019/02/20. doi: 10.1016/j.cmpb.2019.02.007. PubMed PMID: 30777619. for discharge year 2016). We also focus on the comparison of the measures and to show that ICD-10 codes can be used as a predictor for mortality at admission.

Nevertheless, we do agree that some bias will appear in our models and, as such we add an extra paragraph in the Discussion section:

The results of the models may be slightly biased due to repeated measures (e.g. if a patient was admitted more than one time during the study period), as all admissions are treated equally. To overcome this bias, we should remove all admissions previous to the readmissions. In our dataset only 2.5% of the admissions are readmitted within 30 days.

Minor comments;

1. Please add p values to Table 1.

Response: Upon request we added the p-values to Table 1. All variables are, due to the large dataset, significant. 

Table 1 Population overview: characteristics of survivors and non-survivors 

 Population overview

 survivors 

(N: 33 752=97%) non-survivors 

(N: 919=3%) p-value*

Age 52 [30 - 67] 70 [58 - 80] <0.001

Sex ( % male) 17 320 (51%) 543 (59%) <0.001

% Diagnoses Present on admission (PoA) 93% 80% <0.001

DNR at admission 469 (1.5%) 231 (25%) <0.001

Palliative care flag at admission 234 (0.7%) 286 (31%) <0.001

CCI Admission 0 [0 - 2] 3 [1 - 6] <0.001

 Discharge 0 [0 - 2] 3 [1 - 6] <0.001

RoM 

(1 - 2 - 3 - 4) Admission 71% - 22% - 6% - 1% 10% - 34% - 40% - 16% <0.001

 Discharge 70% - 22% - 7% - 1% 6% - 22% - 40% - 33% <0.001

Data are reported as n (%) or medians (1st – 3rd quartile), or otherwise indicated.

* p-values based on Pearson chi-square for categorical variables and the Wilcoxon rank-sum test for continuous variables.

Legend: DNR = Do Not Resuscitate; PoA= Present on Admission flag; CCI = Charlson Comorbidity Index; RoM = Risk of Mortality

Reviewer #2

There are many studies on mortality prediction in the hospital, and this study is very interesting. The number of cases is 30,000 or more, which is a large-scale data, which is very useful in data analysis. In addition, it is novel that the analysis excludes DNR cases, and we believe that this study can provide scientifically important information.

Response: We thank the reviewer for the kind words and for the time spent to improve our study.

---

## [Editor Report · Decision Letter 1]

10 Jun 2020

Using structured pathology data to predict hospital-wide mortality at admission

PONE-D-20-08968R1

Dear Dr. Deschepper,

We’re pleased to inform you that your manuscript has been judged scientifically suitable for publication and will be formally accepted for publication once it meets all outstanding technical requirements.

Kind regards,

Juan F. Orueta, MD, PhD

Academic Editor

PLOS ONE
---

## [Editor Report · Acceptance letter]

11 Jun 2020

PONE-D-20-08968R1 

Using structured pathology data to predict hospital-wide mortality at admission 

Dear Dr. Deschepper:

I'm pleased to inform you that your manuscript has been deemed suitable for publication in PLOS ONE. Congratulations! Your manuscript is now with our production department. 

Kind regards, 

on behalf of

Dr. Juan F. Orueta 

Academic Editor

PLOS ONE